# Critical Computational Evidence Regarding the Long-Standing Controversy over the Main Electrophilic Species in Hypochlorous Acid Solution

**DOI:** 10.3390/molecules27061843

**Published:** 2022-03-11

**Authors:** Ke-Wei Chen, Yun-Dong Wu, Tian-Yu Sun

**Affiliations:** 1State Key Laboratory of Chemical Oncogenomics, School of Chemical Biology and Biotechnology, Peking University Shenzhen Graduate School, Shenzhen 518055, China; ke-wei_chen@stu.pku.edu.cn (K.-W.C.); wuyd@pkusz.edu.cn (Y.-D.W.); 2Institute of Chemical Biology, Shenzhen Bay Laboratory, Shenzhen 518132, China

**Keywords:** electrophilic addition mechanism, hypochlorous acid solution, Cl_2_O, Cl_2_

## Abstract

Although hypochlorous acid (HOCl) solution has become a popular electrophilic reagent for industrial uses, the question of which molecule (HOCl or Cl_2_) undergoes electrophilic addition with olefins remains a controversial issue in some literature and textbooks, and this problem has been largely underexplored in theoretical studies. In this work, we computationally studied the electrophilic addition mechanism of olefins using three experimentally predicted effective electrophilic chlorinating agents, i.e., HOCl, Cl_2_, and Cl_2_O molecules. Our results demonstrate that Cl_2_ and Cl_2_O are the main electrophilic agents in HOCl solution, whereas the HOCl molecule cannot be the electrophile since the energy barrier when directly adding HOCl molecule to olefins is too high to overcome and the “anti-Markovnikov” regioselectivity for tri-substituted olefin is not consistent with experiments. Notably, the HOCl molecule prefers to form oxonium ion intermediate with a double bond, rather than the generally believed chlorium ion intermediate. This work could benefit mechanistic studies of critical biological and chemical processes with HOCl solution and may be used to update textbooks.

## 1. Introduction

Hypochlorous acid (HOCl) solution plays a significant role in physiological processes [1,2,3], pharmaceutical industries [4,5,6], chemical industries [7,8,9,10,11,12], and chemistry education [13,14,15,16,17,18,19] (see Figure 1). HOCl solution is a powerful oxidant that can remove the invading bacteria and pathogens in cells, and it can also undergo electrophilic substitution or addition reactions with biological macromolecules in vivo due to its high electrophilic activity [1,20,21,22,23,24,25], leading to tissue damage and diseases, such as lysosomal lysis, mitochondrial permeability, protease inactivation, and cell necrosis (see Figure 1a) [26,27,28,29]. Moreover, drugs or biological macromolecules containing electron-rich groups can form chlorohydrins and chloramines mediated by HOCl solution [4,5,6], and previous works reported that many diseases may occur through the toxicity caused by the chlorination products of these compounds (see Figure 1b) [4,6]. Furthermore, adding HOCl solution to olefins to synthesize chlorohydrins has recently aroused great interest in industrial production [7,8], e.g., Sargent and co-workers applied this strategy to the electrosynthesis of ethylene and propylene oxides (see Figure 1c) [10]. Finally, HOCl is often selected as a classic case in textbook chapters on electrophilic addition (see Figure 1d) [13,14,15,16,17,18,19].

In a typical industrial plant, the reaction chamber is filled with Cl_2_ gas and water simultaneously, such that HOCl and Cl_2_ molecules are present in a certain proportion in the mixture [11,12]. Therefore, for an electrophilic addition reaction with olefins mediated by HOCl solution, the mechanism cannot exclude the direct addition of Cl_2_ to olefins. Except water as the solvent, this reaction can also occur in various organic solvents such as tert-Butanol (*t*-BuOH) in the Pinnick reaction [30,31]. As shown in Figure 2, there are two plausible and well-established reaction mechanisms that form chlorohydrin. 

To the best of our knowledge, it is widely accepted that the HOCl molecule represents the active electrophile in solutions. A typical example is that proposed by Smith, M. B. et al. on page 996 of Chapter 15 of their eighth edition of *March’s Advanced Organic Chemistry (2020)*, in which the HOCl molecule can be generated in situ through the reaction between Cl_2_ and H_2_O [17], where the Cl group in the HOCl molecule is transferred to the olefins (see Figure 2a). However, McMurry, J. E. et al. proposed the opposite point of view on page 225 of Chapter 8 of his ninth edition of *Organic Chemistry (2015)*, emphasizing that Cl_2_ can directly react with trans-1,2-dimethyl ethylene to form a chlorium ion intermediate, with water as the nucleophilic agent to open the relatively unstable cyclic chloride ring and form a bond with carbon [14]. Furthermore, Rawn, J. D and Ouellette, R. J [16], Bruice, P. Y [15], Loudon, M and Parise, J [18] and Brown, W. H [19] put forward similar pathways in their respective books (see Figure 2b). Therefore, in response to the above contradiction between the two plausible reaction mechanisms, we aim to study which molecule is the active electrophilic species in HOCl solution for double bond addition reactions.

In addition to the two electrophilic species mentioned above, Sivey, J. D. and Roberts, A. L. first proposed that Cl_2_O can be another important electrophilic species in HOCl solution. Their prominent work indicated that the reactivities of the HOCl molecule, Cl_2_, and Cl_2_O were found to be affected by environmental variables, such as the concentration of various effective chlorinating agents and pH [32,33,34]. Therefore, we also performed theoretical calculations on the reactivity of Cl_2_O in this context.

## 2. Materials and Methods

Theoretical calculations were conducted to address the above questions. Two popular density-functional theory (DFT) methods (M06-2X-D3 [35,36], *ω*B97X-D [37]) were used. The triple-ζ augmented correlation-consistent basis sets, designated aug-cc-pVTZ [38,39,40], were employed for all selected atoms. Structures were optimized by Truhlar’s SMD [41] method (solvation model based on the quantum mechanical charge density), with water as the solvent. Frequency analysis was conducted at the same theoretical level to verify the stationary points as an energy minimum or a transition state and obtain the thermal energy corrections. All of the above calculations were performed using the Gaussian 16 program [42]. Furthermore, to confirm the reliability of our DFT results, the highly accurate DLPNO–CCSD(T) method [43,44], which is a domain-based local pair-natural orbital coupled-cluster method, which can perform CCSD(T) [45,46,47] calculations accurately and quickly, was used with a matched basis set extrapolation scheme for single-point electronic energy calculations using the ORCA program (version 4.0) [48,49]. The optimized structure obtained using the M062X-D3/aug-cc-pVTZ theoretical level was used as the initial structure for the DLPNO–CCSD(T) calculation, and the parameters required in the aug-cc-pVDZ and aug-cc-pVTZ two-point extrapolation are from the ORCA manual. In this article, Gibbs free energies are provided for discussion. 

## 3. Results

### 3.1. HOCl Molecule Reaction Pathways

#### 3.1.1. The Cl Group in the HOCl Molecule Adding Pathway

Since many textbooks use trans-1,2-dimethyl ethylene as the electrophilic reactant, it was chosen as the olefin to use to perform the calculations. Firstly, a review of the literature was conducted, but no relevant computational work was found on adding HOCl to olefins, so we inferred that there may be some challenges in this fundamental chemical reaction. Then, we carried out calculations to test the performance of the electrophilic addition of the Cl group in the HOCl molecule to trans-1,2-dimethyl ethylene. Four possible transition states when adding the Cl group in the HOCl molecule to trans-1,2-dimethyl ethylene are shown in Figure 1, involving the most common proposed **TS 1-ClOH** (I); a concerted four-membered transition state for the one-step transfer of the Cl group to one carbon and the simultaneous transfer of the OH group to another carbon (II); a six-membered transition state with one H_2_O molecule as the proton shuttle (III); the asynchronicity transition state (IV). However, none of the four transition states can be located. To verify whether these transition states exist, relaxed potential energy surface (PES) scanning was performed (see Appendix A). Instead of reaching a parabolic peak in the scanning process, the energy of the system continues to increase as the C-Cl bond length is gradually shortened from 2.5 Å to 1.6 Å (the C-Cl bond length in the **Int 1-Cl^+^** is 1.9 Å). Moreover, the chlorium ion **Int 1-Cl^+^** is not stable and is ~15 kcal mol^−1^ higher in energy than the reactants. According to the above computational results, the Cl group in the HOCl molecule adding pathway may not be favorable.

#### 3.1.2. The OH Group in the HOCl Molecule Adding Pathway

It is surprising that, when we tried to locate four-membered transition state II in Figure 1, the OH group transfer transition state (**TS 1-HOCl**) was instead located (Figure 2); its energy barrier is ~30 kcal mol^−1^ relative to reactants by different calculation methods (see Figure 2 and Appendix A); relaxed potential energy surface (PES) scanning also has a peak point similar to the transition state (see Appendix A). The oxonium ion **Int 1-OH^+^** is >32 kcal mol^−1^ more stable than **Int 1-Cl^+^** (see Figure 1 and Figure 2). One water molecule was also used to stabilize the negative charge of the Cl group in **TS 1-HOCl/H_2_O**, but the energy barrier is ~6 kcal mol^−1^ higher than that of **TS 1-HOCl**. 

Considering that **TS 1-HOCl** can be located but **TS 1-ClOH**; **Int 1-OH^+^** is more stable than **Int 1-Cl^+^**, we infer that the OH group transfer pathway is more favorable than the Cl group transfer pathway. However, it is known that the ~30 kcal mol^−1^ energy barrier of the OH group transfer pathway is too high to overcome at the industrial production temperature (35–50 °C) [50], and this electrophilic addition reaction can occur at a very low temperature with a high reaction rate [51,52].

In addition to the high energy barrier, another important issue for the OH group transfer pathway is that, for tri-substituted olefins such as trimethylethylene (see Figure 3 and Appendix A), this pathway leads to an “anti-Markovnikov” product, which is not consistent with the experimental regioselectivity [53]. Therefore, although the transition state could be found, the OH group in the HOCl molecule adding pathway is also not favorable.

The bond-dissociation energy (BDE) of the HO-Cl bond to HO• and Cl• is also high (~60 kcal mol^−1^), and there are no radical initiators in the reaction conditions. Therefore, the radical addition pathway can also be excluded (see Appendix A for more details). The above computational results indicate that the HOCl molecule might not be the electrophilic species to use to carry out the addition reaction with trans-1,2-dimethyl ethylene.

#### 3.1.3. Oxonium Ion Intermediate vs. Chlorium Ion Intermediate

It is generally believed that, in the HOCl molecule, the Cl group is *δ^+^* and the OH is *δ^-^* in terms of electronegativity and that Cl^+^ prefers to form the chlorium ion intermediate with a double bond [54]. However, our calculations do not support this conclusion, and one should not infer the properties of the ions from the molecule since the covalent bond changes their electronic structure. As shown in Figure 4, **Int 1-OH^+^** is >32 kcal mol^−1^ more stable than **Int 1-Cl^+^**. Electronic configuration analysis was used to explain why **Int 1-OH^+^** is more stable than **Int 1-Cl^+^**. The electronic configurations of OH^+^ and Cl^+^ are 1s^2^ 2s^2^ 2p^4^ and 1s^2^ 2s^2^ 2p^6^ 3s^2^ 3p^4^, respectively. There are more electrons in Cl^+^ than in OH^+^ that can shield the positive charge. Therefore, the interaction between the OH^+^ and the double bond is stronger than that of the Cl^+^.

#### 3.1.4. Other Hypohalous Acids (HOF, HOBr, and HOI)

In addition to test the HOCl molecule, we also tested the electrophilic addition of other hypohalous acids (HOF, HOBr, and HOI) to olefins (see Appendix A). Figure 5 shows that the Gibbs free energy of **TS 1-HOX**, **Int 1-OH^+^**, and **Int 1-X^+^** decreases as the electrophilicity of **X^+^** increases. **Int 1-F^+^**, **Int 1-Br^+^**, and **Int 1-I^+^** are all higher in energy than **Int 1-OH^+^**, and **TS 1-XOH** cannot be located, which is in accordance with the HOCl. The energy barriers of **TS 1-HOBr** and **TS 1-HOI** are higher than that of **TS 1-HOCl**, whereas the energy barrier of HOF is achievable (19.6 kcal mol^−1^). Accordingly, the HOF molecule can undergo electrophilic addition with olefin [55]. Notably, a large number of studies have shown that the HOBr molecule is not a good brominating agent in water treatment compared to Br_2_ and Br_2_O [56], which is also consistent with our computations.

### 3.2. The Cl_2_ Reaction Pathway

Next, the reactivity of adding Cl_2_ to olefins was evaluated using the same calculation methods. Our computational results indicate that Cl_2_ is a non-negligible electrophilic chlorinating agent for trans-1,2-dimethyl ethylene. Changes in calculation methods had no discernible influence on the low energy barriers of **TS 1-Cl_2_** (see Figure 6). The energy barrier of **TS 1-Cl_2_** is only 10.5 kcal mol^−1^ according to the high-level DLPNO–CCSD(T) method, indicating that trans-1,2-dimethyl ethylene could be easily chlorinated by Cl_2_, which is in agreement with McMurry’s proposal (see Appendix A).

### 3.3. Another Active Electrophile: The Cl_2_O Reaction Pathway

Considering the complexity of the chemical composition of HOCl solution, the effects of other active electrophiles should not be overlooked. Previous experimental studies have shown that the main chlorinating agents in HOCl solution are affected by many factors, such as pH, chloride ion [32,33], and chlorite concentration [57]. In recent decades, evidence from multiple reports has quantified the influence of electrophilic species in HOCl solution, including HOCl, Cl_2_O, Cl_2_, and H_2_OCl^+^ [58]. Collette, T. W. et al. have denied that reactions between H_2_OCl^+^ and organic compounds are likely [59]. As mentioned above, Sivey, J. D. and Roberts, A. L. conducted remarkable research on the active components of free available chlorine (FAC). They determined the main electrophilic species and their robust rate constants using kinetic experiments and concluded that Cl_2_O and Cl_2_ might be potent electrophilic agents in HOCl solution [32,33,34]. Therefore, the performance of the electrophilic addition of Cl_2_O to trans-1,2-dimethyl ethylene was tested computationally in the present study.

As shown in Figure 7, when the Cl group in Cl_2_O serves as a chlorinating agent to attack trans-1,2-dimethyl ethylene, the energy barrier of **TS 1-Cl_2_O** is only 9.2 kcal/mol, which is lower than that of **TS 1-Cl_2_** using the same calculation methods. Therefore, Cl_2_O and Cl_2_ are the main contributors of olefins in the electrophilic addition reaction rather than the HOCl molecule (see Appendix A). The ability of Cl_2_ and Cl_2_O to function as electrophiles can also be attributed to the superiority of Cl^−^ and ClO^−^ as leaving groups [33].

### 3.4. The Whole PES for the Formation of Chlorohydrins

After determining that Cl_2_ and Cl_2_O are the main electrophilic agents in HOCl solution, we further investigated the following hydrolyzation step to form the final chlorohydrin product (see Figure 8). **Int 1-Cl^+^** is hydrolyzed by water to afford chlorohydrin via **TS 2-Cl^+^-H_2_O**. One water molecule attacks the carbon in the double bond activated with a Cl^+^ ion, and another water molecule is needed to stabilize the leaving proton. The energy barrier of **TS 2-Cl^+^-H_2_O** is reasonable for these reaction conditions.

## 4. Conclusions

In summary, to solve the controversy regarding the active electrophilic addition species (i.e., whether it is the HOCl molecule or Cl_2_) in HOCl solution for the double bond addition reaction, various theoretical calculations were carried out. Our results indicate that the HOCl molecule cannot be added to olefin directly due to its high energy barrier (~30 kcal mol^−1^) and the presence of the wrong regioselectivity. Additionally, electronic configuration analysis was employed to explain the stability of the oxonium ion and chlorium ion intermediates. It is noteworthy that Int 1-OH^+^ is >32 kcal mol^−1^ more stable than the generally believed Int 1-Cl^+^. In comparison, the energy barrier of Cl_2_ attacking the double bond is just ~10 kcal mol^−1^, which makes it a non-negligible electrophilic chlorinating agent for olefin. Moreover, Cl_2_O could become another important chlorinating agent due to its lower energy barrier. Cl_2_O and Cl_2_ are more potent electrophilic agents than the HOCl molecule, which agrees with the earlier assessment by Sivey, J. D. and Roberts, A. L. By and large, this theoretical research unambiguously solves the controversy in some studies and textbooks, and should benefit the mechanistic studies of important biological and chemical processes with HOCl solution.

## Data Availability

More details about DFT calculation are in Supporting Information. This material is available free of charge via the Internet at http://pubs.acs.org.

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
