# Peer review of "Critical Computational Evidence Regarding the Long-Standing Controversy over the Main Electrophilic Species in Hypochlorous Acid Solution"

_molecules, 2022, doi:10.3390/molecules27061843_

Round 1

Reviewer 1 Report

The manuscript "Critical Computational Evidence for the Long-Term Controversy over the Main Electrophilic Species in Hypochlorous Acid Solution" is a very interesting try to put an end in the slong-lasting discussion about the mechanism of olephins chlorohydroxylation. Authors successively analyze different possibilities, try to locate the transition state, and evaluate the activation barrier. They show convinsingly that HOCl itself is hardly an electrophile attacking agent as the transition state either cannot be found or have too high excess energy or attacking agent cannot be formed as the radical reaction initiator is missing. On the other side, Cl2 or Cl2O presenting in the industrial solutions may be such the agents.

The only thing that can be improved, in my opinion: Authors could show how the adduct of olephin with Cl2 formed is hydrolyzed by water. What if the activation barrier of this stage is too high? Probably, there are some literature data available?

I would also appreciate some English polishing: there are some fragments of text that requires struggle to comprehend.

Reviewer 2 Report

In this work, Chen et al. investigated the mechanism of electrophilic addition of HOCl, Cl2 and Cl2O to olefins, using as model case trans-1,2-dimethyl ethylene. The authors claim that there is an apparent controversy in which of the three reactants is the active electrophile in aqueous solutions of hypochlorous acid. To answer this question, they perform a series of quantum chemistry calculations, using the DFT method with different functionals. The major conclusions of their work are: (i)  Cl2 and Cl2O are better electrophilic agents than HOCl  and they claim and (ii) the heterolytic dissociation of the HO-Cl bond leads to HO+ and Cl- and not to Cl+ and HO-.

In my opinion, this manuscript should be rejected due to the following reasons, that I will detail below: (i) the research question is not relevant and the authors disregard obvious alternative hypotheses, (ii) their conclusions are not properly supported by their results and they are hardly meaningful, (iii) their results lack chemical sense and they seemingly suffer from some methodological issue and (iv) the manuscript is poorly written and, in general, of low quality.

I develop the previous points in more detail here:

(i)  The research question is not relevant and the authors disregard obvious alternative hypotheses. The authors claim that there is a controversy in which is the active electrophile in aqueous solutions of hypochlorous acid (HOCl, Cl2 or Cl2O), because they found in different organic chemistry books three possible reactions that lead to electrophilic addition of the chloro group in olefines.  Rather than assuming that only one is possible and calling it a controversy, a much more plausible hypothesis to me is that all three are possible. If you have a solution where all three reactants are present, the more likely scenario is that all three reactions happen and then the real question in which proportion. 

(ii) Their conclusions are not properly supported by their results and they are hardly meaningful. Based on their results, the authors conclude that Cl2 and Cl2O are better electrophilic agents than HOCl because their electrophilic addition reaction has  a lower energy barrier. This hardly answers their supposed controversy since it only means that all reactions are possible except for HOCl, which is basically the null hypothesis. A secondary conclusion of the manuscript is that the heterolytic dissociation of the HO-Cl bond leads to HO+ and Cl- and not to Cl+ and HO-. This conclusion is seemingly surprising for the authors because of the partial charge densities of the groups -OH (negative) and -Cl (positive) in HOCl. In my opinion, such conclusion is hardly surprising since it can be explained by basic chemistry arguments: (i) one should not infer the properties of the ions from the ones of the molecule, since the covalent bond changes their electronic structure, and (ii) it can be simply explained counting the electrons and assigning their electronic configuration, following the two reactions proposed by the authors in page 5 the products are:

  1. Cl+ [Ne] 3s2 3p4  and OH- [Ar] 4s1
  2. Cl- [Ne] 3s2 3p6 and OH+ [He] 2s2 3p5

Reaction 2 is the only one leading to a product with stable electronic configuration (Cl-) thus it is not surprising that it has a lower dissociation energy than reaction 1. 

(iii) Their results lack chemical sense and they seemingly suffer from some methodological issue. The results of the authors consistently show that their intermediary products (aliphatic cations) have lower energies than their reactant (an olefin). I find such results hard to believe since aliphatic cations are known to be highly reactive species while olefins are usually much stable. Thus it is more plausible that olefins have a lower energy than their corresponding aliphatic cation unlike what the results of the authors suggest. It is unclear to me which specific computational procedure the authors used to estimate the energy of each product. The “scan of total energy” that the authors present in the supporting information suggests that their results might suffer from some artifact, such as numerical divergences in the energy minimization, since there are suspicious peaks in the energy profile.

(iv)  the manuscript is poorly written and, in general, of low quality. There are several broken sentences throughout the manuscript which I consider that are not worth listing. Moreover, the authors do not provide any justification of their methodological approach or any explanation of the procedure used to calculate the energetic profiles. There are some unfortunate phrasing issues such as claiming that some results were obtained by serendipity or in other words by chance, which make the reader skeptical of the overall soundness of the research.

Reviewer 3 Report

The manuscript by Sun and co-authors deals with theoretical studies of the mechanism of oxidations with HOCl. Confirmation of the active oxidation species in the aqueous solutions of HOCl is quite interesting problem of inorganic chemistry. However, unfortunately, this work has several serious drawbacks which do not allow me to recommend the publication.

First, I am not happy with the fact that there was not found any TS for the Cl transfer from HOCl. The authors make a conclusion that this pathway does not exist because they could not find the TS. But this conclusion is not convincing. There is always a risk that the TS was overlooked or not found because a flat potential energy surface.

What about another possible transition state for the Cl transfer from HOCl? For instance, a 4-membered TS for the one-step transfer of Cl to one carbon and simultaneous OH transfer to another carbon? Or probably even more favorable 6-membered TS which involves the H2O molecule as a proton shuttle? Finally, can a TS like TS-tri (Figure S3b) exist for the Cl transfer from HOCl? These possibilities should be verified.

Second, the authors try to analyse the effective atomic charges on the Cl and O atoms by calculations of the Cl-O heterolytic bond energies. In fact, these bond energies may have no relation with the effective atomic charges. If they want to analyse the charges why not to calculate them using any of the common schemes?

Third, some calculated values of the energies are not reliable, for instance, the bond dissociation energies for the eqns 1 and 2 and DeltaG values for the formation of int1-Cl+ and int1-OH+. In fact, the SMD implicit solvent model usually fails when numbers of species with the same charge are not equal at the left and right sides of a chemical equation. In this case, the explicit consideration of the solvation is usually necessary (Chem. Comm. 2008, 3930). Therefore, the conclusion about the preferrable dissociation of HOCl to Cl- and OH+ is not solid.

Finally, the general topic of this work may be interesting for the specialists in inorganic chemistry but the scope is not sufficiently broad for the general audience of Molecules.

Minor issue: the “WB97XD” functional. The first letter should be Greek omega but not W.

Rejection is recommended.

Reviewer 4 Report

The manuscript by Sun et al. analyses, from a merely theoretical point of view, the possible interactions of HOCl and olefins, in particular dimethyl ethylene to test the possible mechanism of reaction. Using DFT methods with up to date functionals and extended basis sets, and through comparison also with CCSD calculations, it has been possible to locate a transition state corresponding to the OH group of HOCl attacking the olefin, while no transition state could be located for electrophilic addition of Chlorine. The same  is confirmed for other halogen atoms. In any case, the addition reaction of HOCl-olefin presents too high a barrier. Opposite to this situation, reaction paths for electrophilic addition of Cl2 and Cl2O are energetically possible and well characterized with the same theoretical methods.

Minor point

It would be interesting to show the reaction path, through a relaxed scan, also for some of the possible reaction for which the transition state has been determined. In any case, why in figure S1a-d are there some points outside the traced curve? Because of convergence difficulties?

Round 2

Reviewer 2 Report

I would like to thank the authors for their detailed response to my comments and their efforts to improve their manuscript accordingly. I have revised the new version of the manuscript and I conclude that my main concerns have been properly addressed and I am therefore now positive about its publication.

In my opinion, the manuscript can almost be published as it is and I only have one minor comment left. In the conclusion, the authors state

"Moreover, Cl2O could become another important chlorinating agent due
to its lower energy barrier. Cl2O and Cl2 are more potent electrophilic agents, which agrees with the earlier assessment by Smith, M. B., Sivey, J. D. and Roberts, A. L., and their reactivities are affected by environmental variables such as the concentration of various effective chlorinating agents and pH."

In this sentence, it is implied that the reactivity of Cl2O and Cl2 is affected by the concentration of other chlorating agents and pH of the solution. While I belive that this statement is probably true,  it is phrased in a way that is unclear to me if  (i) it is a conclusion of this study or (ii) an earlier result from the other authors cited in the sentence.

I belive that the intention of the authors was simply to link their work with the existing literature (thus case ii was intented), since their calculations can not properly account for factors such as  concentration of chlorating agents and pH. In that case, I suggest the authors to either include a short description about it in the introduction or rephrase the sentence to make clear that it is a reference to previous work and not a conclusion of this study.

Author Response

Response to Reviewer 2 Comments

Point 1: I would like to thank the authors for their detailed response to my comments and their efforts to improve their manuscript accordingly. I have revised the new version of the manuscript and I conclude that my main concerns have been properly addressed and I am therefore now positive about its publication.

Response 1: We are very thankful that you give us the chance to publish this work. You are so kind. Our research is only from computational aspect and we believe more people will be interested in the experimental work.

Point 2: In my opinion, the manuscript can almost be published as it is and I only have one minor comment left. In the conclusion, the authors state

"Moreover, Cl2O could become another important chlorinating agent due
to its lower energy barrier. Cl2O and Cl2 are more potent electrophilic agents, which agrees with the earlier assessment by Smith, M. B., Sivey, J. D. and Roberts, A. L., and their reactivities are affected by environmental variables such as the concentration of various effective chlorinating agents and pH."

In this sentence, it is implied that the reactivity of Cl2O and Cl2 is affected by the concentration of other chlorating agents and pH of the solution. While I belive that this statement is probably true,  it is phrased in a way that is unclear to me if  (i) it is a conclusion of this study or (ii) an earlier result from the other authors cited in the sentence.

I belive that the intention of the authors was simply to link their work with the existing literature (thus case ii was intented), since their calculations can not properly account for factors such as  concentration of chlorating agents and pH. In that case, I suggest the authors to either include a short description about it in the introduction or rephrase the sentence to make clear that it is a reference to previous work and not a conclusion of this study.

Response 2: We thank the reviewer for pointing out this important scientific issue. You are right that we were simply to link our work with Sivey, J. D. and Roberts, A. L.’s. As you suggested, we have added the following paragraph to the Introduction section, and rephrase the sentence in the Conclusion section to make clear that it is a reference to the remarkable experimental work (marked in yellow).

Introduction: page 2 line 67-72

“In addition to the two electrophilic species mentioned above, Sivey, J. D. and Roberts, A. L. first proposed that Cl2O can be another important electrophilic species in HOCl solution. Their prominent work indicated that the reactivities of HOCl molecule, Cl2, and Cl2O were found to be affected by environmental variables such as the concentration of various effective chlorinating agents and pH.[32-34] Therefore, we also performed theoretical calculations on the reactivity of Cl2O in this context.”

Conclusion: page 9 line 250-251

 “Cl2O and Cl2 are more potent electrophilic agents than HOCl molecule, which agrees with the earlier assessment by Sivey, J. D. and Roberts, A. L., and their reactivities are affected by environmental variables such as the concentration of various effective chlorinating agents and pH.

Reviewer 3 Report

In the revised version, the authors addressed some of my comments. However, there is still one serious issue which should be resolved before this work could be accepted. In their reply, the authors wrote: "An attempt has been made to add explicit water molecules to calculate DeltaG values, but with the addition of explicit water molecules, Int 1-Cl+ and Int 1-OH+ will be attacked by water molecules and decompose." But this means that in the aqueous solution (for which the calculations were carried out) these intermediates do not exist. They are decomposed by water (that is not a surprise for me). It has no sense to discuss the mechanism of a reaction considering intermediates which are not formed in reality. This part of the manuscript should be reconsidered or, if additional calculations of a realistic model will not be successful, deleted.
